# An Indicator System for Evaluating Operation and Maintenance Management of Mega Infrastructure Projects in China

**DOI:** 10.3390/ijerph17249589

**Published:** 2020-12-21

**Authors:** Dan Chen, Pengcheng Xiang, Fuyuan Jia, Jian Zhang, Zhaowen Liu

**Affiliations:** 1School of Management Science & Real Estate, Chongqing University, Chongqing 400045, China; 20180301001@cqu.edu.cn (D.C.); 20180301006@cqu.edu.cn (F.J.); jason2016.cqu@foxmail.com (J.Z.); 2International Research Center for Sustainable Built Environment, Chongqing University, Chongqing 400045, China; 3Construction Economics and Management Research Center, Chongqing University, Chongqing 400045, China; 4Faculty of Civil Engineering and Geosciences, Delft University of Technology, 2628CN Delft, The Netherlands; z.liu-8@tudelft.nl

**Keywords:** mega infrastructure projects, operation and maintenance management, assessment level, indicator system

## Abstract

Mega infrastructure projects provide a basic guarantee for social development, economic construction, and livelihood improvement. Their operation and maintenance (O&M) management are of great significance for the smooth operation and the realization of the value created by the projects. In order to provide an approach for effectively evaluating O&M management, this study develops a holistic indicator system using a mixed-review method from the national macro perspective in China. In this study, literature analysis, policy texts, expert interviews, and grounded theory were used to collect relevant data at home and abroad, and establish an initial evaluation indicator system with 23 indicators covering two dimensions and five aspects. Then the questionnaire survey and factor analysis were used to score and categorize the indicators, and finally an evaluation indicator system for O&M management of mega infrastructure projects was formed. The results show that social relations, environmental benefits, macro policy, and operational capacities play an important role in the evaluation of the O&M of mega infrastructure projects. This study helps the management team to avoid negative impacts in the O&M management of mega infrastructure projects and lays a theoretical foundation for future research. The indicator system in this study is based on the Chinese context, and it remains to be verified whether the indicator system is applicable to other countries due to the differences in political and cultural backgrounds in different regions.

## 1. Introduction

At present, megaprojects all over the world have entered the “trillions era” [1]. During the past few decades, China has carried out the largest infrastructure activities in history, which has become a powerful driving force to promote China’s urbanization construction, economic development, and improvement of people’s living standards [2]. By the end of 2019, the country’s infrastructure investment exceeded CNY 18.5 trillion, the total operating mileage of high-speed railways exceeded 35,000 km, accounting for more than two thirds of the total mileage of high-speed rail in the world, and the mileage of expressways exceeded 140,000 km, ranking first all over the world [3]. Compared with general projects, mega infrastructure projects, as large-scale social activities, involve a wide range of scope and construction difficulty, consume a lot of capital, and have significant complexity in technology, environment, organization, and management [4]. This kind of project has higher requirements for decision-making, design, construction, and operation and maintenance (O&M). Once the project has problems, not only the project itself, but also the economy, environment, and society closely related to the project will cause great harm [5]. According to the data from the Gartner Group, 80% of the lifecycle cost of projects occurs in the O&M management phase, and 40% of the projects never resume operation again [6]. At the same time, the impact of the O&M of mega infrastructure projects on society and the environment has also been widely a cause for concern for academics. For example, the Three Gorges Dam, the world’s largest hydroelectric project and a symbol of China’s confidence in high-risk technological solutions, has been heavily criticized for the threats it poses to the environment, animal species, and migrants [7,8,9]. The report by McKinsey shows that by 2030, the investment in mega infrastructure projects will reach $ 57 trillion, which means that the annual investment will reach $ 6-9 trillion, accounting for 8% of the global GDP [10]. In the context of global sustainable development, the sustainable development of mega infrastructure projects has become a hot topic in academia.

Issues related to the O&M management of mega infrastructure projects have been a concern from different perspectives. Researchers in the organization and management field have emphasized the importance of O&M management throughout the lifecycle management (LM). These studies cover a variety of topics, such as input–output of funds [11], resource consumption [12], and satisfaction of all participants [13]. In the field of project management, the complexity, ambiguity [14], politics [15], risk, and ambition [16] of mega infrastructure projects have been explored. It also involves the management of stakeholders from different levels [17]. The long-term impact of O&M management on society, the economy, and the natural environment is also considered along with the measurement of the distribution of benefits [18]. In addition, economists and policy experts described the macro impact of the projects. These issues are related to economic growth [19], labor market [20], regional health [21], social stability [22], market integration [23], and national productivity [24]. In practice, the O&M management of mega infrastructure projects mainly relies on the experiences and intuition of managers, lacks a theoretical basis, and the conclusions are relatively subjective. The indicator system in published studies is usually based on the project and is used to measure performance [14] and technical [25] and teamwork capabilities [20], and these indicators can only meet the general project requirements. Therefore, there is an urgent need for a complete indicator system to quantitatively assess the O&M management of mega infrastructure projects.

Nowadays, the evaluation of the O&M of infrastructure projects mostly focuses on transportation infrastructure. The indicators are only for one or some types of infrastructure [19,23], and some only measure one aspect of the infrastructure, such as environmental [26,27] or economic factors [28]. There are few indicators considering the macro impact of O&M on the government and society, which cannot systematically and comprehensively evaluate the O&M of mega infrastructure projects. Based on this, this study uses a combination of quantitative and qualitative analysis to construct the indicators for the O&M of mega infrastructure projects so as to lay a theoretical foundation for the quantitative analysis of O&M management.

The indicator system constructed in this paper is tailored to the O&M of mega infrastructure projects in the Chinese context. These indicators can effectively evaluate the O&M of mega infrastructure projects and pave the way for quantifying them. The structure of this study is as follows: The second section provides a literature review on the O&M of mega infrastructure projects. The third section introduces the research methodology. The fourth part describes the process of the indicator system. Last, the fifth section summarizes the findings and future direction of the research, and the contributions and limitations of this study.

## 2. Literature Review 

Since the 1990s, as an important part of LM, O&M management has been a hot topic in academia. O&M management generally involves space management, public safety management, asset management, and energy consumption management. Because of its long duration and large cost, O&M management plays an important role in LM. Previous academic studies are mostly found in the field of power and telecommunication engineering to evaluate and discuss the cost control, resource scheduling, and efficiency evaluation of projects [25]. In the field of engineering, O&M management research mainly focuses on the application, innovation, and deficiencies of BIM (Building Information Modeling) [26], the Internet of Things, 3D printing [27] and other technologies in the O&M management of projects. 

However, the O&M management of mega infrastructure projects is different from that of general projects. The U.S. Presidential Commission for Critical Infrastructure Protection (PCCIP) [28] made an authoritative definition of critical infrastructure projects in its report: Critical infrastructure refers to the facilities that can weaken a country’s national defense strength and have a significant impact on economic security after failure or damage. Eight key infrastructures are defined in the report: communication system, power system, oil and gas, banking and finance, transportation system, water supply system, government service, and emergency service system. The object of this study was mega infrastructure projects, which have the following characteristics: (1) They are key projects in national or local government economic development planning; (2) they concern strategic and public welfare; (3) they have far-reaching impacts on the national or regional economy, society, and ecological environment; (4) they are of great concern to the public. They are not a combination of general projects, and the purpose of them is often to change the world [14]. The O&M management of this kind of project has a great impact on society, the economy, and the ecological environment, and may cause social chaos and inequality in certain areas [25]. In the context of global economic integration, mega infrastructure projects have become a research hotspot because of changing production and lifestyle, and have brought a huge and permanent impact to the Earth [1]. In terms of the management of mega infrastructure projects, scholars have conducted exploratory studies from different perspectives. From a corporate perspective, scholars have focused on aspects such as green construction [29], building safety [30], environmental management [31], and public pressure [32]. From a project point view, the O&M management of mega infrastructure projects that span multiple geographic regions is also a key part of project management. In the process of O&M, it is proposed to strengthen the links between the central government and local governments, local governments at all levels, and social organizations and citizens [33] so as to achieve a win-win situation for all parties through the joint participation of multiple subjects [34]. From a social point of view, the impact of mega infrastructure projects goes far beyond the projects themselves—for example, the impact of the Qinghai–Tibet Railway on the ecological aspects of the Tibetan region and the impact on the living standards and lifestyles of the residents of the Tibetan Plateau. There is evidence that the construction and O&M of mega infrastructure projects can enhance national productivity, promote regional economic development, and create employment opportunities for the local people [35]. Establishing indicators for the O&M management of mega infrastructure projects is not suitable due to the complexity and impact of the operation process. First, existing indicators are based on the project perspective and examine O&M in terms of quality, cost, schedule, and safety [36], which cannot reflect the social benefits and project sustainability. Second, some indicators originate from the enterprise perspective, which only focuses on the input–output of the enterprise and specific aspects of the enterprise’s resources [37], benefits, and services [38]. Many scholars, experts, and institutions have researched and constructed various types of indicator systems, ranging from social benefits to economic benefits to a comprehensive system of indicators for assessing the usefulness of projects [39]. The existing indicators and criteria are only applicable to enterprises in the construction industry, and in such an indicator system, the social and ecological impact of the operation is excluded from consideration. Third is a system of indicators from the micro level. This kind of study usually covers a category of infrastructure projects, such as highway [19] and railway [40], which are limited in their indicators and cannot reflect the overall O&M management of mega infrastructure projects. Moreover, these fragmented indicators are difficult to aggregate due to different perspectives. Although the current sustainability initiatives, strategies, frameworks, and processes for the O&M management of mega infrastructure projects emphasize broader national aspirations and strategic goals [41], there are still serious challenges in integrating these national strategic objectives (such as the economy, society, and the environment) into the micro level. Given the current state of the research, there is an urgent need for methods and techniques to evaluate the O&M management of mega infrastructure projects, and to assess making decisions about unexpected problems in the process.

In recent years, scholars have used various methods to construct evaluation indicators for different research objects. Commonly used methods include literature review [42], questionnaire survey [43], case analysis [44], semi-structured interview [45], the Delphi method [46], multi-criteria decision-making [47], and grounded theory [48]. Based on the complexity of the projects, the methods used in this study include literature analysis, policy texts, expert interviews, grounded theory, questionnaire survey, and factor analysis to comprehensively and systematically sort out, summarize, and filter the O&M management indicators of mega infrastructure projects so as to build a complete, effective, and comprehensive indicator system.

## 3. Methodology

This study aims to filter the indicator system and clarify the evaluation indicators in the O&M management of mega infrastructure projects. To end this, this article adopts the “mixed-review method” [49,50]. Generally speaking, this method includes qualitative analysis (i.e., literature analysis, policy texts, etc.) and quantitative analysis (i.e., questionnaire survey and factor analysis) so it can eliminate biased conclusions and subjective interpretations while providing an in-depth understanding of domain knowledge.

The mixed-review method refers to the combination of quantitative analysis and qualitative analysis. This method can not only reduce the subjective judgment influence brought by qualitative analysis, but also deepen the understanding by quantitative analysis so as to make up for the shortcomings of a single method. As a practical application of mixed-method research, the mixed-review method combines the quantitative evaluation method and qualitative review method in the process [51]. It can reduce the impact of subjective judgment of manual qualitative review methods, and improve the depth and understanding of the results of quantitative study methods [52]. This study takes literature analysis, policy texts, and grounded theory as the qualitative analysis, and questionnaire survey and factor analysis as the quantitative analysis. The mixed-review method is divided into three stages (see Figure 1). The first stage is to collect the indicators for evaluation of the O&M of mega infrastructure projects, and then sort out the relevant indicators through literature analysis, policy texts, and expert interviews in this stage. During this stage, we can comprehensively understand the research in this field, and collect, identify, and organize previous studies from all aspects. The second stage is the identification of indicators, in which the information collected in the first stage is sorted out and the preliminary list of indicators for measuring the O&M management of mega infrastructure projects is identified by combining it with the grounded theory. The purpose of this stage is to summarize, analyze, and evaluate the research subject. The third stage is the construction of the indicator system, which will determine the key indicators of the O&M management of mega infrastructure projects through questionnaire survey and factor analysis, and systematically construct the final evaluation indicator system of the O&M management of mega infrastructure projects.

### 3.1. Step 1. Indicator Collection

Literature review is a method of collecting, identifying, and organizing previous studies to form a scientific understanding of the target domain [42]. It is the most basic method that provides a valuable theoretical basis for in-depth analysis in the field of the research. In addition, due to the uniqueness of mega infrastructure projects, the government (including the central government and local government) has the absolute right in LM [53], so it is necessary to organize and summarize the policies, systems, laws, and other documents issued by the government to analyze the influence of the government on the O&M management of mega infrastructure projects from the macro level. Indicator collection is crucial in this study, so the academic literature and policy texts must be carefully selected. As shown in Figure 1, the data acquisition channel (i.e., the selection of academic literature and policy texts) is first selected for the analysis below. (1) Academic literature: With the help of keywords such as “large-scale project,” “large-scale infrastructure,” “complex project,” “megaproject,” “operation and maintenance management,” “indicator,” and “evaluation” in the core collection of Google Scholar, Web of Science, CNKI (China National Knowledge Infrastructure), and school digital library, study was carried out in the databases of the science core collection. In order to cover the literature in the research field as comprehensively as possible, a subject search selected to define the publication time range was defined as 2010–2020. (2) Policy text: Central and local government networks, official websites such as the NDRC (National Development and Reform Commission) and the Ministry of Housing and Urban–Rural Development, documentary reports of mega infrastructure projects, official websites, special websites, news reports, and related sustainability reports and initiatives.

In addition, as the world has entered a new phase of globalization, an increasing number of megaprojects have attracted extensive attention from scholars. Universities and institutions are enhancing their research on megaprojects by establishing independent research centers. Therefore, in addition to academic literature and policy texts, this study complements the constructed indicators for evaluating the O&M management of mega infrastructure projects through expert interviews. The expert interviews can directly obtain valuable interview data through face-to-face interaction with the interviewees, which is helpful for clarifying ideas and identifying problems [54]. In particular, the expert interview process divided into two parts. (1) Interviews with experts in practice. Interviews were conducted with eight practice experts (sample size determined according to the principle of theoretical saturation) who had participated in mega infrastructure projects [45], and their views on project O&M management were solicited from the interviewees as a supplement to the initial indicators, taking into account the actual situation of specific megaprojects. The interviewed experts had a range of 5–30 years of experience in project management of megaprojects, including middle and senior management of the three owners, two constructors, and three project management consultants (two of whom are also part-time staff at research institutions) (the background information of the interviewed experts is in Appendix B). (2) Interviews with academic experts. The compiled initial indicators were discussed with academic experts to solicit their opinions on the reasonableness and scientific validity of the setting and presentation of the indicators in the initial scale and to determine the initial indicators for the evaluation of the O&M management of mega infrastructure projects. All three academic experts were senior professors and doctoral supervisors in the field of engineering management with rich experience in both research and practice.

### 3.2. Step 2. Indicator Identification

The indicator system needed to include systematicality, validity, independence, scientificity, and operability. In order to ensure the applicability of the indicators and the comprehensive evaluation of mega infrastructure project O&M management, we used the grounded theory to analyze the elements of the indicator system. Grounded theory is a qualitative analysis method that leads the heory by combing the development of a phenomenon on the basis of collecting a large amount of data [55]. Based on the literature and the situation of this study, the main steps were theoretical sampling, data coding (including open coding, principal axis coding, and selective coding), theoretical saturation testing, and results [48]. The theoretical sampling was based on the literature analysis, policy texts, and the records of eight expert interviews. The data coding was based on the results of the literature and texts. The theoretical saturation test was based on the expert interview records (Figure 2). So as to establish relationships between conceptual classes to construct an initial indicator system for the O&M management of mega infrastructure projects, laying the groundwork for the final indicator system was the next stage.

### 3.3. Step 3. Establish Indicator System 

Based on the results of first two steps, the questionnaire was administered under the theme of “evaluation of the O&M management of mega infrastructure projects.” In order to achieve the research objectives, the questionnaire was divided into two parts: The first part was the basic information survey, including the respondents’ background information (i.e., personnel statistics) and organization information (basic information of the project). The second part focused on the evaluation of the O&M management of mega infrastructure projects. The questionnaire was designed with the Likert scale, and assigned 1–5 points according to the degrees of “totally disagree,“ “disagree,” “neutral,” “agree,“ and “totally agree,” respectively [45]. In order to ensure the reliability of the questionnaire, on the one hand it was necessary to carefully consider the way to express the questions in the questionnaire, which needed to be clear, objective, and easy to understand. On the other hand, the questionnaire did not clearly reflect the content and logic of the study, so as to prevent the interviewees from getting the possible implication of a causal relationship and affecting the questionnaire responses. The questionnaire was sent to engineers and managers with mega infrastructure project experience, including contractors, supervisors, and operators (Appendix A for details of the questionnaire). They were asked to rate each indicator in relation to their own work experience. In order to improve the authenticity and validity of the questionnaire, we collected the questionnaires through both online and face-to-face distribution. Finally, a total of 197 questionnaires were collected and 184 were valid, with an effective rate of 93.4%. T recovery rate exceeded 80%, which was statistically significant. [45].

Considering that there could be strong correlation between some indicators of the O&M management of mega infrastructure projects, the 23 initial indicators identified above were classified by the factor analysis method. The categories of indicators that have a significant impact on the O&M management of mega infrastructure projects were summarized and the indicators were classified according to the metric weight so as to establish the indicator system and provides a basis for the evaluation of the O&M management of mega infrastructure projects, which follows up on research on the O&M management of mega infrastructure projects.

## 4. Results and Discussion

### 4.1. Collection of Evaluation Indicators

#### 4.1.1. Indicator Collection Based on Literature Analysis

In order to develop a suitable and comprehensive indicator system for evaluating the O&M management of mega infrastructure projects, the first stage was to identify the aspects. A review of the literature revealed that there were few empirical studies on the O&M of projects, and even fewer from the perspective of project management, specifically focusing on the O&M management of mega infrastructure projects. Therefore, the O&M indicator system studied in this paper is essentially a complete system set up to achieve the ultimate goal of the operating entity of mega infrastructure projects—that is, the level of O&M is identified by judging the extent to which mega infrastructure projects achieve their operational goals. Studies have shown that the O&M of mega infrastructure projects can have an obvious and significant impact on the economy, society, and environment. The O&M of mega infrastructure projects can be judged from both internal and external aspects, and the indicators were extracted and summarized to obtain the indicator system shown in Table 1.

#### 4.1.2. Indicator Collection Based on Policy Texts

Most of the mega infrastructure projects are invested and constructed by the government, which plays a leading role in the LM of the projects. This ensures the successful realization of their functional, economic, social, and environmental benefits [66]. Therefore, the construction and O&M management of mega infrastructure projects are also the reflection of government policies and systems. Based on this, it is necessary to evaluate the O&M management of mega infrastructure projects from the political dimension to reflect the uniqueness. Since there are few studies that deal with the political aspects and it is impossible to obtain the influencing factors objectively through literature analysis, this study researched policy texts by collecting publicly available textual materials of mega infrastructure projects, such as from the State Council’s policy document library, “China South-to-North Water Diversion Network,” the official website of Gezhouba Company, the main operator of the Three Gorges Project, the official website of the Hong Kong–Zhuhai–Macau Bridge Authority, the news feature “The Full Completion of the Qinghai–Tibet Railway,” and in news features, newspapers, and other channels. By sorting out news reports, policy documents, case studies, and relevant reviews on the O&M management of mega infrastructure projects, nearly 50,000 words of textual materials were compiled. Through keyword induction and extraction, the political indicators of the O&M management of mega infrastructure projects were finally obtained. The detailed results are shown in Table 2.

#### 4.1.3. Indicator Collection Based on Expert Interviews

This study used a combination of interviews with practicing experts and academic experts to explore the factors influencing the O&M management of mega infrastructure projects [57]. Most of the expert interviews were conducted with no more than 20 people [46], so eight practice experts who participated in mega infrastructure projects were interviewed and asked for their opinions on the factors as a supplement to the initial indicators as follows: (1) economic aspects. Experts agreed that the impact of mega infrastructure projects on the economy should be analyzed from the perspective of changing the industrial structure of the economy and improving the income level of residents, and experts from government departments also mentioned that the O&M management of mega infrastructure projects contributed to an increase in government financial revenue. The significance of economic structure and economic function was unclear in the evaluation, and was suggested to be deleted. The economic indicators changed significantly, eliminating the secondary indicator and adding three indicators: changing the economic and industrial structure, improving residents’ income level, and increasing the government’s fiscal revenue. (2) Social aspects. Experts believed that the O&M management of mega infrastructure projects in the context of China could be evaluated separately from political and social aspects, and political factors should be mainly analyzed by policy texts, so the original indicator system was split. (3) Technology aspects. Experts believed that the technical level and technical risk were influential factors that enhanced or constrained the O&M management of projects, rather than evaluation indicators used to judge the O&M of mega infrastructure projects, so it was suggested that the whole system be deleted. The technical aspect was therefore deleted. (4) Participant aspects. Experts believed that the O&M management of projects was not only evaluated from the overall benefits to the external environment, but was also an important indicator to evaluate the O&M management of mega infrastructure projects themselves. The satisfaction degree of investors and the public overlapped with the indicators of social and economic aspects, which was suggested to be deleted.

The second round of expert interviews: The indicator system constructed in the first two stages was submitted to three experts in academia for their opinion in order to test the reasonableness of the indicator system and the accuracy of the question formulation on the basic scale. Experts discussed and revised the structural design and terminology of the indicators, and finally reached a consensus as shown in Table 3.

### 4.2. Initial Indicator System

With the assistance of the mentor and team members, the abovementioned channels (3.1) were used to collect relevant information on mega infrastructure projects from June 2020 to August 2020, and 70,000 words of literature were compiled. Through the extensive reading of the literature, a deeper understanding of the concept, scope, and performance of O&M management was obtained, which formed the theoretical knowledge of O&M management as a criterion for text search and target selection. The coding analysis of the materials using NVivo yielded the following results: (1) open coding. After comparing the literature and policy texts, the indicators with the same or similar meanings were combined. A total of 52 tags and 163 initial sentences were sorted out from the documents. The evaluation indicators were extracted from the literature and policy texts, and the existing indicators in the documents were deleted. After repeated reorganization, integration, and refinement, 23 subcategories were obtained. (2) Spindle coding. The 52 labels and 23 subcategories obtained from the open code were carefully compared, and these labels and categories were placed in the context of mega infrastructure projects. Combined with the actual situation in the process of project O&M management, five main categories of evaluation indicators were obtained. (3) Selective coding. This study selected “O&M management of mega infrastructure projects” as the core coding, and adopted the idea of a storyline to guide coding [48]. First of all, combined with the comparative analysis of raw materials and codes at all levels, the story context was defined as the O&M management of mega infrastructure projects evaluated by political, economic, social, environmental, and operational teams. Then, these indicator aspects were taken as the concrete embodiment of the O&M management of mega infrastructure projects by connecting and comparing the main category and the subcategory. Finally, it combed the relationship between the core category and other category levels. After systematic analysis, the core category, the main category, and the subcategory constituted a whole. 

In order to improve the reliability and validity of the findings, a theoretical saturation test was conducted. In-depth interviews were conducted with eight practice experts on the consistency and completeness of the descriptions of the indicators selected in 4.1. Nearly 50,000 words of interview records were compiled, and 13 labels were obtained by coding and analyzing the interview transcripts. Further coding and analysis did not form a new core category and relationship, and no new theory was found in the main category. Therefore, the model integration process was reliable and theoretical saturation was realized. The mega infrastructure projects were constructed through grounded theory. The initial indicator system of O&M management evaluation is shown in Table 4.

### 4.3. Screen Key Indicators 

#### 4.3.1. Descriptive Statistics

Descriptive statistical analysis is a statistical method that uses mathematical language to describe sample characteristics or explain the relationship between variables [43]. A total of 184 valid samples were statistically analyzed, and the results are shown in Table 5. The number of male and female respondents in the survey was 148 and 36, respectively, with a ratio close to 4:1, which is consistent with the industry characteristics in that there are more men than women in the projects. From the perspective of an education background, half of the respondents had bachelor’s degrees, and the proportion of master’s degrees and doctor’s degrees accounted for 26.09% and 19.57%, respectively, which indicates that the educational background of those working in the construction industry is constantly increasing. A total of 50% of the respondents had been involved in the O&M management of mega infrastructure projects for 6–10 years. Only 6.52% of the respondents had more than 10 years of work experience, indicating that the O&M management of mega infrastructure projects is still in its infancy. A total of 100 of the respondents had been involved in the O&M management of highways, 36 in the O&M of bridges, 24 in the O&M of railways, 16 had participated in the O&M of water conservancy projects, and only 8 respondents had been involved in the O&M of power grid projects, which reflects the ratio of different types of projects in mega infrastructure to some extent. Due to the particularity of the project, the respondents were located in research institutes, government agencies, and operation management departments. The units issued by the questionnaire included most of the participants in O&M management, which was a good way to achieve multi-party participation and ensure the quality of the questionnaire. The fact that almost all of these parties belonged to the state indicates that the operation process of mega infrastructure projects is under the supervision of the government. The distribution of these data basically reflects the actual proportion of researchers in mega infrastructure projects in China. These proportions basically reflect the actual distribution of O&M management in China. The data also suggest that the respondents were experienced and knowledgeable about the issues under this study, which increased the confidence in the data quality.

#### 4.3.2. Factor Analysis

Factor analysis is a process of identifying relatively few factor groups, which is used to represent the relationship between multiple groups of interrelated variables [67]. In order to confirm the applicability of the data collected from the questionnaire survey, two tests were needed: (1) the sample size and the strength of the relationship between the indicators [68], and (2) the number of samples collected being at least five times the number of indicator variables [69]. A total of 184 valid questionnaires were used to rate 23 indicators. The sample size was eight times the indicator variable to meet the demand. The valid questionnaire results were imported into SPSS 23.0 (IBM corporation, Stanford, CA, USA), and the data were standardized. The KMO (Kaiser-Meyer-Olkin) value was 0.816 (>0.7) [69], and the significance level of the Bartlett spherical test was <0.0001 [70], as shown in Table 6, which shows that the sample size was sufficient and the correlation coefficient matrix indicates strong correlation between the indicators, which could be used for factor analysis. The common degree of indicators was close to or greater than 0.8, indicating that the common factors had a strong explanatory power for variables and were suitable for factor analysis. The results of the KMO and Bartlett spherical test indicate that the initial indicators for evaluating the O&M management of mega infrastructure projects have good construction validity.

According to the standard extraction factor, eigenvalues reater than 1 when the characteristic value is less than 1 [71], and the total variance interpretation is shown in Table 7. It can be seen that a total of four common factors were extracted from the 23 evaluation indicators, and the cumulative total variance explained by the eigenroots of the factor was 74.439%, which is greater than 60% [72], indicating that the extracted common factors can effectively reflect the O&M management of mega infrastructure projects.

Since the typical representative variables of each common factor in the initial factor solution were not prominent and not convenient for the analysis of the actual problems, for this consideration, the rotation was carried out using the maximum variance method to obtain the well-defined common factors. After four iterative convergence cycles, the rotated factor load matrix was obtained, and the 23 indicators were reduced to four common factors, as shown in Table 8 and Table 9.

Common factor 1 included indicators A_1_, A_2_, A_4_, A_5_, A_6,_ B_1_, C_1_, C_2_, C_4_, and C_5_, which were five political indicators, four indicators for social aspects, and one indicator for economic aspects. These indicators all reflected the impact of mega infrastructure projects on social relations, so they were named “social relations.” Common factor 2 included indicators C_3_, D_1_, D_2_, D_3_, and D_4_, namely, four environmental indicators and one social indicator. These indicators reflected environmental benefits, so they were named “environmental benefits.” Common factor 3 included A_3_, B_2,_ B_3_, B_4,_ and B_5_, namely, four economic indicators and one political indicator, which reflected the impact of project O&M management on macro policy, so they were called “macro policy.” Common factor 4 included E_1_, E_2_, and E_3_, which was completely consistent with the indicators of operation capacity, so the name “operation capacity” was retained.

#### 4.3.3. Results Discussion

Through factor analysis, four factors of O&M management of mega infrastructure projects were concluded, which were social relationship, environmental benefit, macro policy, and operation capability. According to the indicator contribution rate ranking, it can be seen that social relations play a crucial role in the O&M management of mega infrastructure projects. The indicators of each factor are described in detail below.

##### Factor 1: Social Relations

The social relations factor included 10 indicators: consistent with policies, legal and strategic approaches, meeting the requirements of national defense construction, adjustment of policy differences related to the project, coordinating of social contradictions among regions related to the project, narrowing of regional economic differentials in relation to the project, changing the economic and industrial structure, maintaining social stability, meeting the needs of social development, improving employment, and making full use of social resources. The essence of any infrastructure project is to serve society [73], and China is no exception. In social relations, respondents generally believed that the O&M management of mega infrastructure projects had a high score on maintaining community relations, meeting the needs of social development, and improving regional employment, with an average score of more than 4, indicating that the O&M management of mega infrastructure projects played an important role in promoting these three aspects. The ratings were lower for coordinating policy differences and reducing economic disparities between regions related to the projects. It shows that the role of mega infrastructure projects in reducing regional political and economic disparities has not been fully exploited. Mega infrastructure projects generally span two or more provincial administrative regions [74]. There are also differences in economic level and guidelines in the policies for project O&M management between different regions. The O&M of mega infrastructure projects requires unified standards and collaborative governance across different provinces to maximize social benefits and better serve society. During the interview, a policy advisor from the Transport Bureau said that the core purpose of the construction, delivery, and operation of mega infrastructure projects was to ensure social stability. The central government and local governments at all levels are committed to increasing the proportion of government spending on infrastructure projects so as to ensure the quality and avoid the emergence of “jerry-built” projects, which have a significant negative impact on society. Academic experts added that the O&M management of mega infrastructure projects plays a crucial role in improving the resilience of society as a whole.

##### Factor 2: Environmental Benefits

The second factor environmental benefits consisted of five indicators: significant social benefits, meeting the needs of environmental protection, protecting the regional resources, making full use of local resources, and having a positive impact on the local ecological balance. Respondents generally gave low scores to environmental benefits, with the lowest score among the 23 indicators being for satisfying environmental needs, conserving local resources, and adequate and rational use of local resources. Respondents generally believed that the O&M management of mega infrastructure projects has little effect on the protection of resources and environment. More than half of the suggestions in the self-selected section of the questionnaire indicated that at this stage, all stakeholders are trying their best to ensure that the completion and the O&M management of the project does not damage the surrounding natural environment and ecological balance, but the positive impact on the surrounding environment is almost non-existent. Respondents also indicated that the environmental protection problems in the O&M management of mega infrastructure projects have not been given sufficient attention, and the green operation concept has not been well implemented. Interviewees from NGOs (non-governmental organizations) said that throughout the LM of mega infrastructure projects, there is a certain degree of impact on the ecological environment, and the vast majority of the impact is negative. University experts said that infrastructure should adhere to the concept of green and low-carbon from the design, construction, operation, and final demolition in the process of project delivery, energy conservation, water conservation, rational use of resources, and other contents. Government representatives said that in recent years, the concept of green, low-carbon, and sustainable development has indeed been paid attention to in the construction industry, and has also been applied throughout the whole lifecycle management of megaprojects. The negative impact of projects on the surrounding environment has gradually decreased. Moreover, governments at all levels have special funds to support the introduction of advanced green technologies from abroad.

##### Factor 3: Macro policy

The third type of factor, macro policy, included five indicators: strong sustainability, improving residents’ income level, government revenue increasing, promoting regional GDP growth, and meeting the tax target of the project. In terms of macro policy, the respondents generally believed that the O&M management of mega infrastructure projects has a positive effect on the sustainable development of society and the improvement of economic level. Among the indicators, the highest score of strong sustainability was 4.02, which indicates that the sustainability of mega infrastructure project operation receives recognition, and almost all respondents believed that the O&M management of mega infrastructure project contributes to regional development, and vice versa. The scores of each indicator show that the economic benefits of the project are considerable and have been recognized by the public. As the respondents from the government said, local governments generally hold a positive attitude towards the construction of mega infrastructure projects. In addition to being in line with the national macro policy, local governments can also accelerate the development of regional industries such as tourism, increase employment opportunities, and enhance the regional economy as a whole. A university professor said that the construction of mega infrastructure project requires the local government to take a stand and weigh the realities of the region and deliver mega infrastructure projects without considering the actual needs, which may bring heavy economic burden to the local government and people.

##### Factor 4: Operation Capability

The fourth factor, operational capacity, consisted of three indicators: education level of staff, experiences of staff, and the number of practitioners participating in projects. It was not difficult to find that the average age of mega infrastructure project O&M management personnel is 35.31 years old, and the project operation time is generally between 1–5 years. The frontline and grassroots operation staff are mainly young employees. Although the entire project lifecycle is planned and designed as early as the planning period, the professional quality of the operation team is also needed to deal with the unexpected situations that arise during the actual operation. Mega infrastructure projects, especially transportation infrastructure, cause huge losses and social influence when accidents occur. Data shows that the role of staff in the O&M of transport infrastructure is much greater than 26% [75], so improving the skills of staff in monitoring and handling malfunctions is an effective way to ensure the safe and smooth operation of infrastructure. According to the research of safety expert Heinrich, there is an “88:10:2” rule in the operation safety of infrastructure [75]. That is, in 100 safety accidents, 88 are caused by human causes, 10 are caused by people and objects, and only 2 are caused by “natural disasters” that are difficult to prevent. Both human factors and equipment factors have to be resolved by the staff. Therefore, it is very important to continuously improve the ability of employees to perform their duties and the rational allocation of the skill structure of the staff team, and to improve the team’s ability to deal with failures. An official from the transportation bureau said that they recruit employees openly every year and train them intensively on a regular basis. University representatives said that to improve the professional level of the operations team, not only should we consider the matching of people and positions, but also the rationalization of the team’s professional skill structure. Optimized team structure and reasonable manpower allocation are conducive to improving the operational efficiency of infrastructure safety.

### 4.4. Construction of Indicator System

The results of the factor analysis show that the initial indicator system of the O&M management of the mega infrastructure projects has a reasonable category of indicators, and some of the categories can be further refined. The indicators were ranked according to the influence on O&M management, and the final evaluation indicator system for the O&M management of mega infrastructure projects is shown in Table 10.

## 5. Conclusions

### 5.1. Findings and Contribution

To fuel rapid economic growth, a number of impressive mega infrastructure projects have been built all over the world. These planned mega infrastructure projects are unprecedented, and they have exacted a great influence on economic construction, social development, and people’s livelihood. However, it should also be noted that this construction takes a toll on local communities and the natural environment. The consequences have gone beyond the project itself and have evolved into a series of social problems because of lack of O&M management. In this context, it is important to evaluate the O&M management of mega infrastructure projects and identify the problems in the operation process. Based on the macro perspective, this study determined 23 initial indicators of the O&M management of mega infrastructure project from two dimensions and five aspects by using literature analysis, policy texts, expert interviews, and grounded theory, and evaluated the relative importance of indicators from the participants’ perspective through questionnaire survey. Finally, 23 indicators were grouped into four factors by factor analysis, namely, social relations, environmental benefits, macro policy, and operation capacity, which resulted in an indicator system for evaluating the O&M management of mega infrastructure projects.

This study provides contributions in three ways. First, the indicator system of O&M management of mega infrastructure projects aims to provide a basis for the evaluation and decision-making of the O&M management of megaprojects. For these projects, the operation directly determines the performance of the project and the impact on society and the economy. Paying attention to the O&M management of megaprojects helps to improve the operation performance. Second, the indicator system not only includes the social, economic, and environmental aspects of the previous research, but also innovatively includes the political aspects, which is different from the general project operated for economic benefit. The O&M management of mega infrastructure projects is aimed at serving society [76]. By evaluating the role of project operation in adjusting the disparity of political, economic, and social development in different regions to judge the role of the project in national macro control, not only can we clearly judge the operation performance of the project, but also promote the sustainable development of the project. Finally, this study adopts a quantitative approach to measure the impact of project O&M management on different indicators by calculating the mean score of respondents for different indicators. This method can visually reflect the intuitive feedback of the respondents and overcome the obstacles of considering different levels of issues, which helps to objectively reflect the real sense of the respondents.

### 5.2. Limitations and Future Directions

This study has several limitations, but also provides several promising areas for future research. First of all, our indicator system was established based on the context of China. The O&M management models adopted by different countries and regions are significantly different due to differences in regional economic development and cultural backgrounds. Generalization must be validated in the future to find out whether our indicator system can be appropriate for general engineering projects and for other countries around the world. Second, it is important to acknowledge that there are contextual factors that may affect the presence and scoring of these O&M management indicators. For example, national circumstances, cultural characteristics, and different types of projects could greatly influence O&M management (highway, railway, water conservancy, power grid, and bridge). An in-depth discussion of these problems is beyond the scope of this study. Furthermore, because the context is constantly changing, the indicator system should be dynamically adjusted according to the time and the national environment in which it is applied. 

This study provides new ideas for the field of project O&M management and lays a theoretical foundation for the quantitative analysis of the O&M management of mega infrastructure projects. In the next stage, we will focus on the evaluation and improvement of the O&M management of mega infrastructure projects and carry out empirical studies for one or some types of mega infrastructure projects.

## Figures and Tables

**Figure 1 ijerph-17-09589-f001:**
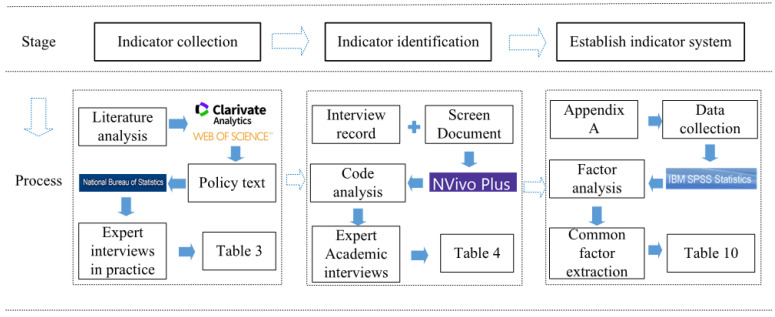
Research framework and process.

**Figure 2 ijerph-17-09589-f002:**
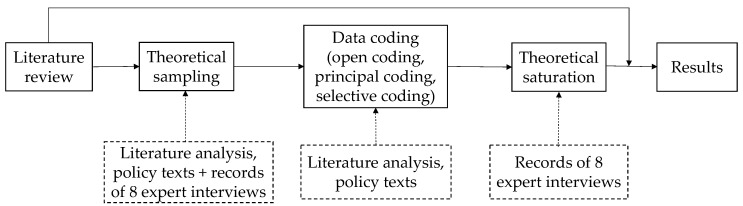
Process of Grounded theory.

**Table 1 ijerph-17-09589-t001:** Indicators from studies.

Aspect	Indicator	Related Literature
Economy	Resource allocation	Ahola et al. (2014) [50]
	Resource exploitation and utilization	Li et al. (2016) [51]
	Economic exchange	Tauterat (2015) [38]
	Industrial distribution	Pollack et al. (2017) [15]
	Industrial structure	Mok et al. (2017) [17]
	Consumption structure	Li et al. (2018) [52]
	Labor structure	Qi et al. (2016) [32]
	Incomes	Chen et al. (2014) [53]
	Government revenue	Bernardo (2014) [49]
	Impact on GDP	Biesenthal et al. (2014) [54]
Social/Political	Policy guarantee	Li (2010) [55]
	Legal protection	Ma et al. (2006) [16]
	Rules and regulations	Xue et al. (2015) [27]
	Maintaining social stability	Matten (2008) [56]
	Meet social needs	McWilliams (2001) [57]
	Social benefits	Peloza (2009) [58]
	Social resource	Zeng et al. (2015) [24]
Technology	Quality performance	Ogata (2015) [37]
	Technology innovation	Lin et al. (2014) [29]
	Technology maturity	Tidd (2017) [41]
	Technical difficulties	Davies (2009) [59]
	Technical requirement	Bstieler (2015) [60]
Environment	Ecological protection level	Levitt (2007) [14]
	Ecological impact	Lin et al. (2015) [31]
	Resource assurance level	Du (2015) [39]
	Resource utilization	Peng (2007) [40]
Participant	Payback period	Mok et al. (2017) [13]
	Investment yield period	Moodley (2008) [18]
	Happy operation	Daily et al. (2003) [61]
	Reasonable profit margin	Zhai (2009) [62]
	Meeting the scheduled tax	Aguinis (2012) [63]
	Meet the special needs	Aguinis (2012) [63]
	Satisfaction with the project	Li et al. (2016) [64]
	Acceptance of the project	Li et al. (2015) [65]

**Table 2 ijerph-17-09589-t002:** Indicators from policy texts.

Aspect	Indicator
Politics	Consistency with policy, legal, and strategic approaches
	Meeting the requirements of national defense construction
	Strong sustainability
	Adjustment of policy differences related to the project
	Coordination of social contradictions among regions related to the project
	Narrowing of regional economic differentials in relation to the project

**Table 3 ijerph-17-09589-t003:** Indicators from expert interviews.

**Dimension**	**Aspect**	**Indicator**
Operational Benefits	Politics	Consistency with policy, legal, and strategic approaches
		Meeting the requirements of national defense construction
		Strong sustainability
		Adjustment of policy differences related to the project
		Coordination of social contradictions among regions related to the project
		Narrowing of regional economic differentials in relation to the project
	Economy	Changing the economic industrial structure
		Improving residents’ income level
		Government revenue increasing
	Social	Maintaining social stability
		Significant social benefits
		Making full use of social resources
		Meeting the needs of social development
	Environment	Meeting the needs of environmental protection
		Protecting the regional resources
		Making full use of local resources
		Having a positive impact on the local ecological balance
Operational Capability	Operation Team	Education level of staff
		Experiences of staff
		Number of practitioners participating in projects

**Table 4 ijerph-17-09589-t004:** Initial indicator system.

Dimension	Aspect	Indicator
Operational Benefits	Politics (A)	Consistency with policy, legal, and strategic approaches (A_1_)
		Meeting the requirements of national defense construction (A_2_)
		Strong sustainability (A_3_)
		Adjustment of policy differences related to the project (A_4_)
		Coordination of social contradictions among regions related to the project (A_5_)
		Narrowing of regional economic differentials in relation to the project (A_6_)
	Economy (B)	Changing the economic industrial structure (B_1_)
		Improving residents’ income level (B_2_)
		Government revenue increasing (B_3_)
		Promoting regional GDP growth (B_4_)
		Meeting the tax target of the project (B_5_)
	Social (C)	Maintaining social stability (C_1_)
		Meeting the needs of social development (C_2_)
		Significant social benefits (C_3_)
		Improving employment (C_4_)
		Making full use of social resources (C_5_)
	Environment (D)	Meeting the needs of environmental protection (D_1_)
		Protecting the regional resources (D_2_)
		Making full use of local resources (D_3_)
		Having a positive impact on the local ecological balance (D_4_)
Operational Capability	Operation Team (E)	Education level of staff (E_1_)
		Experiences of staff (E_2_)
		Number of practitioners participating in projects (E_3_)

**Table 5 ijerph-17-09589-t005:** Summary of the statistics of the interviewees.

Indicators	Number	Proportion
Male	148	80.43%
Female	36	19.57%
Average age	35.31	
Education		
PhD	36	19.57%
Master’s	48	26.09%
Bachelor’s	92	50%
Other	8	4.35%
Work experience		
≤5 years	80	43.48%
6–10 years	92	50%
>10 years	12	6.52%
Job position		
Top manager	26	14.13%
Middle manager	48	26.09%
Other	110	59.78%
Type		
Highway	100	54.35%
Railway	24	13.04%
Water conservancy	16	8.7%
Power grid	8	4.35%
Bridge	36	19.57%
Institute		
Government	48	26.09%
Research institutes	64	34.8%
Operating unit	72	39.11%

**Table 6 ijerph-17-09589-t006:** KMO and Bartlett spherical test.

KMO Sampling Appropriateness		0.816
Bartlett Spherical Test	The approximate chi-square	1080.214
	Degree of freedom	253
	Significant	0.000

**Table 7 ijerph-17-09589-t007:** Explanation of total variance.

Component	Initial Eigenvalue			Sum of Squares of Rotating Loads		
	Characteristic Value	Variance%	Cumulative%	Characteristic Value	Variance%	Cumulative%
1	12.236	53.200	53.200	12.236	53.200	12.236
2	2.472	10.750	63.950	2.472	10.750	2.472
3	1.365	5.936	69.886	1.365	5.936	1.365
4	1.047	4.553	74.439	1.047	4.553	1.047
5	0.965	4.196	78.439			
6	0.730	3.175	81.810			

**Table 8 ijerph-17-09589-t008:** Rotating posterior factor load matrix.

Indicator	Common Factor 1	Common Factor 2	Common Factor 3	Common Factor 4
A_1_	0.589	0.010	0.179	0.158
A_2_	0.586	0.270	0.391	0.070
A_3_	0.434	0.341	0.542	0.324
A_4_	0.788	0.161	0.256	0.185
A_5_	0.810	0.109	0.144	0.350
A_6_	0.697	0.018	0.362	0.259
B_1_	0.820	0.214	0.120	0.228
B_2_	0.582	0.158	0.588	0.212
B_3_	0.411	0.149	0.821	0.137
B_4_	0.576	0.124	0.671	−0.002
B_5_	0.335	0.252	0.821	0.139
C_1_	0.662	0.554	0.130	0.043
C_2_	0.734	0.324	0.188	−0.076
C_3_	0.502	0.529	0.318	−0.017
C_4_	0.726	0.322	0.249	−0.084
C_5_	0.716	0.497	0.094	0.092
D_1_	0.262	0.843	0.187	0.141
D_2_	0.171	0.833	0.310	0.276
D_3_	0.139	0.783	-0.026	0.287
D_4_	0.210	0.816	0.228	0.290
E_1_	0.015	0.424	0.501	0.514
E_2_	0.127	0.321	0.243	0.728
E_3_	0.324	0.370	0.013	0.784

**Table 9 ijerph-17-09589-t009:** Public factor extraction after dimensionality reduction.

Common Factors	1	2	3	4
**Indicator**	A_1_, A_2_, A_4_, A_5_, A_6_, B_1_, C_1_, C_2_, C_4_, C_5_	C_3_, D_1_, D_2_, D_3_, D_4_	A_3_, B_2_, B_3_, B_4_, B_5_	E_1_, E_2_, E_3_
**Naming**	Social relations	Environmental benefit	Macro policy	Operational capability

**Table 10 ijerph-17-09589-t010:** Final indicator system.

Dimension	Indicator
Social Relations	Consistency with policy, legal, and strategic approaches
	Meeting the requirements of national defense construction
	Adjustment of policy differences related to the project
	Coordination of social contradictions among regions related to the project
	Narrowing of regional economic differentials in relation to the project
	Changing the economic industrial structure
	Maintaining social stability
	Meeting the needs of social development
	Improving employment
	Making full use of social resources
Environmental Benefits	Significant social benefits
	Meeting the needs of environmental protection
	Protecting the regional resources
	Making full use of local resources
	Having a positive impact on the local ecological balance
Macro Policy	Strong sustainability
	Improving residents’ income level
	Government revenue increasing
	Promoting regional GDP growth
	Meeting the tax target of the project
Operational Capability	Education level of staff
	Experiences of staff
	Number of practitioners participating in projects

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
