# Peer review of "An Indicator System for Evaluating Operation and Maintenance Management of Mega Infrastructure Projects in China"

_ijerph, 2020, doi:10.3390/ijerph17249589_

Round 1

Reviewer 1 Report

The authors have made efforts to improve the paper based on review comments, while there are still problems regarding the methodology that should be addressed.

  1. In terms of the two-round Delphi method, according to 3.1 and 4.1.3, the authors conducted interviews with 8 industry experts in the first round and then 3 academic experts in the second round for verification and supplements. This is not a Delphi method, which should target the same group of experts and aim to arrive at a group opinion or decision by surveying several rounds.
  2. Figure 1 is not clear and complete. How two rounds of interviews conducted? Why the middle box only includes literature not policy texts and transcripts? How the use of grounded theory be reflected? Where is the questionnaire? What are the meanings of different shapes of text boxes?
  3. For stage 1 and stage 2, what the exact process of literature/policy analysis, interview and NVivo analysis? According to 3.2 and 4.2, the authors mentioned that "the study uses NVivo to code and analyze the primary data such as literature, policy and interview transcripts". In 4.1.3, they stated that "the results of the first round of interviews supplemented and refined the metrics obtained from literature analysis and policy text as follows." In 4.2, they highlighted that "in order to improve the reliability and validity of theoretical research, 8 interview records were coded again and 13 tags were obtained. Further coding and analysis did not form a new core category and relationship, and no new theory was found in the main category." I am quite confused that what is the purpose of interviews? In stage 1, are the interviews based on findings from literature and policy analysis, and is there any coding procedure applied to literature and policy texts? And what are the raw materials included for Nvivo coding? how many documents are included? 

Reviewer 2 Report

The article has been improved a lot with the authors having addressed comments from the reviewers. Things including why this study and the research design are now become clearer. Previous concern over the experts interviewed, the respondents surveyed and the suitability of using questionnaire survey no longer remains. Yet, the authors are suggested to review the language and simplify the sentences in the camera-ready version.

Round 2

Reviewer 1 Report

Thanks for the effort to improve the paper based on the comments. Now clear. 

This manuscript is a resubmission of an earlier submission. The following is a list of the peer review reports and author responses from that submission.

Round 1

Reviewer 1 Report

Although it is not a new topic, the article is consistent and well developed.

It should be noted, as the authors themselves acknowledge, that it is essentially based on Chinese reality, which does not prejudice the article, but recommends that this limitation be mentioned in the abstract and in the introduction.

Reviewer 2 Report

The paper develops an indicator system for evaluating the operation and maintenance management of mega infrastructure projects, which is well written in general. This reviewer has the following comments for authors to further address:

  1. For the developed evaluation indicator system (Table 11), variables for operational capability are neutral while for other three categories are positive, which are not consistent. Besides, possible criteria for these variables to be actually used for evaluation are not provided.
  2. The authors used the terms "indicator" "dimension" "aspect" and "variable" in a mix which is a little bit confused. Are variables in tables 1-4 and table 11 is the ones you mentioned in the conclusion that "23 indicators were grouped into four categories by factor analysis"? whether "Operational capability" is an indicator or dimension?
  3. "Chang the economic industrial structure" in Table 11 is a typing mistake
  4. It would be more clear to separate results and discussion in 2 sections.

Reviewer 3 Report

The article is original and of great interest and responds to the objectives of the journal. The literature review is solid and the methodological design is very detailed and well done. The results obtained have a theoretical and practical dimension.

However, and as the authors themselves point out, the entire article is based on its empirical part, results and conclusions in the Chinese context. This fact in the subject matter, mega infrastructures, means that the results obtained can hardly be applied in other countries for obvious economic and cultural reasons. China has carried out in the last decades, and, indeed, throughout its history, the construction of colossal and unique infrastructure works around the world. Being a communist single-party dictatorship and in a social context in search of extreme economic development, some works have been carried out that in the rest of the world would not have had the same economic support and, above all, political unanimity.

Ultimately, it is essential that the title clearly specify that the study is carried out in China in the way that the authors consider. The reader and researcher who is interested in this article must be clear that it is carried out in the context of China's mega infrastructures.

Reviewer 4 Report

This manuscript can be interesting but I hesitate to recommend acceptance of it at the moment. Below are a few comments and questions to the authors for further consideration:

  • A few fundamental questions: Why develop an indicator system for evaluation of O&M of mega infrastructure project? And how to apply the developed indicators for the intended evaluation and to assist decision making? The authors did point out there is a research gap in this area, but why evaluation of the O&M of the mega infrastructure projects is important and how the developed indicators can assist the decision making?

  • Another fundamental question – what are mega infrastructure projects? The authors should provide a definition so as to minimise ambiguity and to confine the scope of study more clearly.

  • How the developed indicator system can quantitatively assess O&M? (Line 74-75)

  • Line 78 – “the evaluation of the operation and maintenance of infrastructure … focused on transportation infrastructure, and the indicators are single, one-sided…” Any reference and any more details? This point also relates to the development of a generic or specific indicator system mentioned below.

  • The purpose of this study (line 162) is unclear. It is difficult to understand that the authors try to filter the indicator system and clarify the evaluation indicators.

  • The research design is on the complicated side. First, indicator collection through literature review (academic plus policy) and analysis, then expert interviews with those from both industry and academia using Delphi method. Second, grounded theory approach was adopted and NVivo was used to analyse the data to construct an initial indicator system. Third, a questionnaire survey in which respondents were asked to rate the indicators was conducted, followed by factor analysis to summarise the indicator categories for the construction of the final evaluation indicator system. The authors did attempt to explain the rationale behind this design but not in a clear and precise manner, e.g. how did the factor reduction and extraction process helped the development of the final indicator system?

  • As the authors reported, stage 1 is the collection stage while stage 2 is the identification stage. A number of research activities have been carried out in these two stages, including literature analysis, expert interviews and “combination with the grounded theory”. Can the authors elaborate the differences between collection and identification, and why collection and identification are done separately in two stages?

  • Problems are spotted in the questionnaire survey and they must be clarified and improved. What have been done to ensure the validity and reliability of the survey result? A significant portion (about 43.5%) of respondents have less than 3 years of experience in mega project. Their suitability for the questionnaire survey is questioned. Besides, only 30% of the respondents are top or middle manager. The majority (70%) indicated their job position as others. The authors hinted that others are actually those frontline or grassroots staff (line 501-502). Their appropriateness to this questionnaire survey is also questioned. What’s more, why those working in research institutes are considered to be suitable for this questionnaire survey?

  • In stage 1, any details of the “eight practice experts” and “academic experts” while keeping their anonymity?

  • The authors need to explain and elaborate “questionnaire star”. Is it an app that enables online questionnaire survey? Again, how reliability and validity is ensured when the questionnaire survey is conducted using both face-to-face and online questionnaire survey?

  • The current version is not ready to be recommended for publication – (a) the MDPI template lines have not been removed (line 176-178); (b) the language has not been checked thoroughly, mistakes and errors in language are common, e.g. “Previous academic studies have comprehensively and systematically discussion the cost control … “ (line 102-103), “184 valid samples were statistically analysed, and the results are shown in Table 5” (line 357) – Table 5 showed the demographics of the “interviewees”, not the results? and “Participate in the survey of 100 people participated …” (line 366); (c) Missing in-text citation and ambiguous text, e.g. “According to a researcher, data shows that … of transport infrastructure is much greater than 26%, … (line 507-508); and (d) In Part 2 of the Appendix, there are problems in (i) 4. Work Experience, one tick box is 3 years, not less than 3 years? (ii) the corresponding dimension of the variables.

  • As the authors acknowledged, the indicator system is developed in the Chinese context and may not be applicable to other countries (Line 561-563). This point must be made clear in the article title and right from the beginning, that the indicator system is for mega infrastructure projects in China.

  • Clarification of the second highlighted limitation is needed (line 564-570). If the authors are developing a generic indicator system, it is inevitable that “some contextual factors that may affect the existence of indicators as well as the scoring” (line 565). It is rather strange to say this is beyond the scope of this study. Are the authors suggesting to develop different indicator systems to evaluate specific types of mega infrastructure projects?

  • Last but not least, the reviewer also questions whether this manuscript falls within the scope of the journal, particular how the study of O&M indicators relates to health?